# Assessing the metabolic and physiological costs of oviparity in the epaulette shark (*Hemiscyllium ocellatum*)

Carolyn R. Wheeler[1,2,*], Cynthia A. Awruch[3,4], John W. Mandelman[2,5] and Jodie L. Rummer[1,6]

**ABSTRACT**

Reproduction in chondrichthyan fishes (sharks, rays, skates, and chimaeras) is generally assumed to be a long-term, energetically costly process, given their slow generation times. However, metabolic costs of reproduction remain poorly understood due to a lack of direct, non-lethal measurements. To address this, we investigated metabolic and physiological changes during oviparous reproduction in five female epaulette sharks (*Hemiscyllium ocellatum*). We tracked oxygen uptake rates – a proxy for metabolic rate – across a 3-week cycle, capturing data before, during, and after egg case encapsulation and oviposition. We also measured reproductive hormones (testosterone, 17β-estradiol, progesterone) and hematological parameters (hematocrit, hemoglobin concentration). Results revealed a positive but non-significant relationship between metabolic rate and body mass, and contrary to expectations, metabolic rate did not significantly change throughout the 19-day cycle. Hormone levels remained stable, except for a transient testosterone peak early in the cycle, and hematological parameters showed no significant variation. These findings tentatively suggest epaulette sharks maintain reproductive effort without marked increases in metabolic or physiological costs. Continued research under seasonal environmental variation could clarify reproductive energetics in chondrichthyans further. This study provides the first direct measurement of metabolic effects of oviparous reproduction in chondrichthyans, challenging assumptions about energetic demands in this taxon.

**KEY WORDS: Chondrichthyes, Elasmobranch, Metabolic rate, Energetics, Steroid hormones, Breeding strategy**

**INTRODUCTION**

Reproduction is a fundamental biological process essential for population maintenance, requiring significant energetic investment by the individual (Bell 1980; Reznick 1985). Direct parental reproductive energetic investments encompass hepatic and gonadal

[1]ARC Centre of Excellence for Coral Reef Studies, James Cook University, Townsville, QLD 4814, Australia. [2]School for the Environment, The University of Massachusetts Boston, Boston, MA 02125, USA. [3]IMAS (Institute for Marine and Antarctic Studies), University of Tasmania, Hobart, TAS 7001, Australia. [4]CESIMAR (Centro Para el Estudio de Sistemas Marinos) – CENPAT – CONICET, Puerto Madryn, Chubut U9120ACD, Argentina. [5]Anderson Cabot Center for Ocean Life, New England Aquarium, Boston, MA 02110, USA. [6]College of Science and Engineering, James Cook University, Townsville, QLD 4811, Australia.

*Author for correspondence (carolyn.wheeler23@gmail.com)

C.R.W., 0000-0001-9976-8420; C.A.A., 0000-0003-0047-7848; J.W.M., 0000-0002-4679-2694; J.L.R., 0000-0001-6067-5892

processes, such as vitellogenesis leading to folliculogenesis and oogenesis in females, and spermatogenesis in males. Additionally, indirect maternal reproductive efforts involve various physiological and behavioral changes, including alternations in locomotion due to shifts in body morphometrics, pregnancy maintenance, brooding behaviors, and parental care (Smith and Wootton, 1995; Gillooly and Baylis, 1999; Olsson et al., 2000; Vézina et al., 2006; Reardon and Chapman, 2010; Van Dyke and Beaupre, 2011; Foucart et al., 2014; Buddle et al., 2020). The degree of reproductive effort is inherently linked to the reproductive strategy, with viviparity (live-bearing) generally requiring more sustained effort than oviparity (egg-laying) due to the extended duration and maintenance demands associated with gestation (Ginther et al., 2024). However, it is important to note that this temporal difference in effort does not necessarily imply a difference in net lifetime reproductive output (Foucart et al., 2014). In vertebrates such as fish, reptiles and birds, reproductive costs are reflected in shifts in female metabolism for both live-bearing and egg-laying species (DeMarco, 1993; Hopkins et al., 1995; Angilletta and Sears, 2000; Masonjones, 2001; Nilsson and Råberg, 2001; Vézina and Williams, 2002; Vézina et al., 2003, 2006; Munns, 2013; Foucart et al., 2014; Finotto et al., 2023). Nevertheless, comparative data on metabolic changes during reproduction in fishes remain scarce (Arnold et al., 2021), highlighting a critical gap in our understanding of reproductive energetics.

This study quantified fine-scale changes in the metabolic rates associated with reproduction in a chondrichthyan fish (class of fishes including sharks, rays, skates and chimaeras). This group includes approximately 400 oviparous species, which are characterized by their late age of sexual maturity, extended reproductive cycles, and low fecundity (Cailliet et al., 2005). These life-history traits suggest that reproduction could be an energetically demanding process. Despite the diverse reproductive strategies within this taxon, however, there is limited research on the specific energetic costs associated with these modes (Lawson et al., 2019, 2022). Current knowledge is derived from bomb calorimetry of embryos and reproductive tissues (e.g. Lawson et al., 2022), assessments of hepatosomatic and gonadosomatic indices (HSI and GSI, respectively), which reflect energy storage and utilization in relation to reproduction (e.g. Hoffmayer et al., 2006; Awruch et al., 2008), and measurements of circulating energetic blood markers in reproducing individuals (e.g. Hammerschlag et al., 2018; Rangel et al., 2021a,b). However, these approaches may not adequately capture the temporal dynamics of reproductive energetic effort, which requires measurement of whole-organism metabolic changes over time (Van Dyke and Beaupre, 2011). To date, only one study has temporally measured changes in metabolic rate associated with reproduction in a chondrichthyan species, reporting a 78.5±35.2% reduction in metabolic rate in viviparous rays post-parturition (Finotto et al., 2023). No study to our knowledge has explored the metabolic costs of oviparous reproduction in chondrichthyans.

Biology Open

In this study, we used the epaulette shark (*Hemiscyllium ocellatum*), an oviparous, long-tailed carpet shark, as our model species. The reproductive biology of this species is well-documented, both from studies of wild populations at Heron Island on the Great Barrier Reef, Australia (Heupel et al., 1999) and observations of captive animals (West and Carter, 1990; Payne and Rufo, 2012). Epaulette sharks exhibit a single oviparity reproductive mode, producing only one egg case per uterus at any given time, typically forming two egg cases per cycle across their two uteri. Once an egg case is formed and undergoes sclerotization (hardening), it is deposited onto the benthos shortly thereafter, with the entire process spanning only a few days (Nakaya et al., 2020). In the wild, reproduction in epaulette sharks is seasonal, with peak egg-laying occurring from September to December, likely ceasing as summer water temperatures rise (Heupel et al., 1999). In captivity, however, epaulette sharks can produce eggs year-round when maintained at a constant water temperature that simulates the natural laying season (Heupel et al., 1999). Given the extensive knowledge of their reproductive biology and their amenability to captive breeding, epaulette sharks serve as an excellent model for studying the metabolic costs associated with reproduction. Here, we employed a combination of methodologies and metabolic and hormonal markers to gain a detailed understanding of the physiological regulation and energetic costs of reproduction in epaulette sharks.

## RESULTS

### Female reproductive cycle

From the five reproducing individuals in this study, we observed 196 egg cases deposited over 98 cycles from March 2020 to January 2022. The female reproductive cycle length averaged 19 (±5.7 s.d.) days long at 25°C, where intra-individual variability was low (Table 1). Each cycle at 25°C produced a median value of two egg cases, where a few cycles at the beginning of the reproductive season only produced one wind egg case (i.e. an empty egg case with no yolk sac) or three egg cases (two containing yolk-sacs and one wind). Egg case encapsulation on average required 5.3 (±3.2 s.d.) days, ranging from 2-10 days (Table 1). For the majority of cycles assessed, the two egg cases of the clutch were laid between 0-2 nights apart (Table 1). When egg cases were not deposited on the same night, the egg case from the right uterus was deposited first in all cases. Female sharks consumed their full feeding rations throughout the reproductive cycle, except when within 2 days of egg deposition. During this window, in 35% of feedings, sharks did not consume any food, but resumed normal feeding after oviposition occurred.

### Routine metabolic rate ($\dot{M}O_2$)

Respirometry was performed 37 times throughout this study, and there was no evidence of a conditioning effect from the repeated

trials in any of the sharks (see Fig. S1). Furthermore, the small diel temperature cycle in this study (24.5-25.5°C) did not affect $\dot{M}O_2$ (Wheeler et al., 2022). There was no effect of female reproductive status on $\dot{M}O_2$ but there was detectable intra-individual variability in $\dot{M}O_2$ (Figs 2,3, Table S2).

### Hematological parameters

Across female reproduction, T peaked during the pre-encapsulating stage, and both $E_2$ and $P_4$ were consistent across the cycle (Fig. 4, Tables S3–S5). Across all hematological parameters assessed (Hct, [Hb], and MCHC), there were no significant changes (Fig. 5, Tables S6–S8).

## DISCUSSION

In this study we aimed to use fine scale daily tracking of female epaulette shark oviparity to assess effects of this biological process on metabolic demands. We expected that female RMR would increase during egg case encapsulation and then decrease after oviposition, but our findings did not reflect these changes (Figs 2 and 3). A possible explanation of these findings may be related to the lack of seasonal timing and its effect on breeding strategy of the sharks in this study. When epaulette sharks are reproducing seasonally in the wild, they are likely to employ a capital breeding strategy, where energy is stored in the liver during the resting period from January to July. Then, during the reproductive season from August to December, these hepatic energy stores are mobilized via vitellogenesis to create ovarian follicles for reproduction (McBride et al., 2015). However, because epaulette sharks in this study were stimulated via water temperature to reproduce year-round, females may have transitioned to an income breeding strategy to continuously acquire energy via food intake for continuous egg production (McBride et al., 2015). Indeed, unpublished data suggests that, when offered food to satiation, mature females consumed higher percentages of their body weight per feeding than mature males (B. Lewis, unpublished data). Furthermore, given the continuous nature of egg production here and that $E_2$ and $P_4$ hormones did not change across the egg production cycle, vitellogenesis and subsequent folliculogenesis were likely occurring continuously in the background (Koob and Callard, 1999). Upon necropsy of female epaulette sharks, pairs of similarly sized follicles increasing in size have been noted, where follicles for any given cycle have been forming over the course of several months (Wheeler and Rummer, 2024). Therefore, the continuous nature of vitellogenesis and folliculogenesis within an income breeding strategy may keep the general size and therefore oxygen demand of the ovary consistent across time, and therefore no changes in RMR were detected.

One alternative explanation for the lack of RMR variation is that the fasting period prior to respirometry may have suppressed metabolic rate to a baseline maintenance level, temporarily halting

**Table 1. Individual female reproductive cycle data means (±s.d.) of the cycle length (from one set of egg cases to the next), egg case encapsulation time (time the egg cases were present in the oviduct), intra-cycle time (time from oviposition of the first egg case to the second within one cycle), total length (TL), and mass**

| Individual | Cycle length (days) | Egg case encapsulation time (days) | Intra-cycle time between egg cases (days) | TL (cm) | Mass (kg) |
|---|---|---|---|---|---|
| Female 1 | 16.3±3.8 | 7.5±1.4 | 0.5±0.9 | 81.0 | 1.46±0.095 |
| Female 2 | 19.5±5.2 | 6.2±3.1 | 0.8±0.8 | 68.0 | 1.14±0.123 |
| Female 3 | 17.6±5.1 | 3.1±1.2 | 1.5±1.2 | 73.9 | 1.09±0.076 |
| Female 4 | 20.0±3.5 | 5.0±2.6 | 2.1±1.4 | 75.8 | 1.07±0.123 |
| Female 5 | 30.4±6.4 | 8.1±3.2 | 4.8±2.1 | 70.8 | 0.96±0.080 |
| Mean | 19.0±5.7 | 5.3±3.2 | 1.7±1.7 | 73.9±5.0 | 1.16±0.202 |

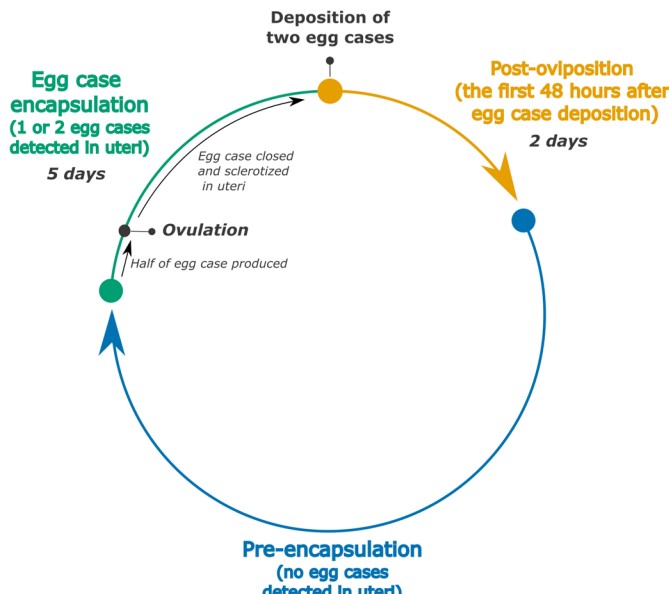

**Fig. 1. Schematic of the categorization of the reproductive cycle of epaulette sharks.** The colors of each phase correspond to all subsequent figures. Days noted for each phase represent the average across all individuals.

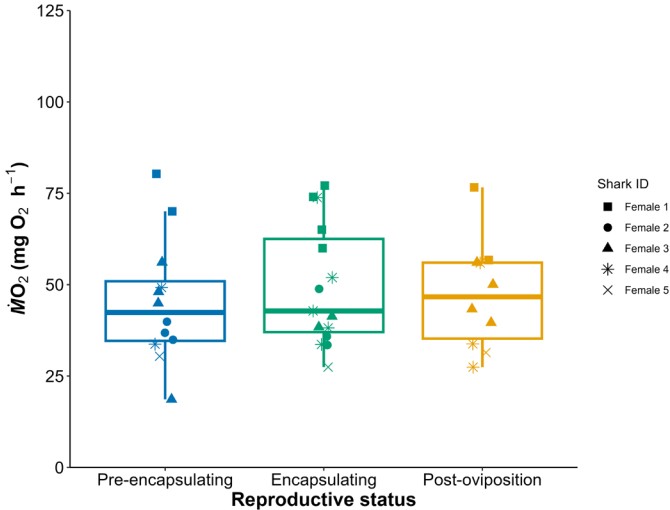

**Fig. 3. Boxplots of metabolic rate [as RMR (mg $O_2$ h$^{-1}$)] across female epaulette shark oviparity.** There were no statistically significance differences between reproductive statuses at $\alpha$=0.05.

energetic investment into reproduction. However, our observations do not support this interpretation. In multiple cases, females were palpated and confirmed to have eggs *in uteri*, and fasting was initiated with the intent of capturing RMR during encapsulation. Despite this, several females deposited their eggs during the fasting period, prior to respirometry, indicating that oviposition proceeded even in the absence of recent feeding. This suggests that short-term fasting did not energetically delay egg deposition, and that the reproductive tract remained active during respirometry preparation. Moreover, reproductive output was monitored throughout the study, including during non-respirometry periods, and no differences in egg production was observed. These findings support the

conclusion that the fasting protocol did not suppress reproductive effort.

Within the egg production cycle, many of the main reproductive changes occur within the oviducal gland, where this gland is responsible for secretion and formation of the egg case capsule and is thought to have a considerable metabolic requirement (Callard et al., 2005). The oviducal gland begins formation of the egg case prior to ovulation of the follicle, so the gland must be controlled by endocrine factors, presumably from the ovary (Callard et al., 2005). Oviducal gland growth coincides with increased circulating $E_2$ in other oviparous elasmobranchs (Koob et al., 1986; Heupel et al., 1999), so in this study where $E_2$ was consistent across the egg production cycle, the oviducal gland may have remained consistent in size across constant reproduction, thereby maintaining RMR. These glands are small in mass compared to the whole shark mass, and therefore any increased oxygen demand of these glands during egg case encapsulation may be minimal in comparison to the oxygen demand of the whole organism. In other words, even though the oviducal gland during encapsulation may have a high metabolic demand, the glands are small, and therefore the increased demand is not well reflected in whole organism oxygen uptake rates. Future research should assess oviparous reproduction using seasonal changes in water temperature to mimic wild conditions that up and down regulate the reproductive tract to help capture the overall energetic effort of reproduction beyond the egg production cycle. Furthermore, altering feeding amount and/or food items whilst tracking egg case production characteristics (i.e. the amount and category – e.g. number of wind cases produced) may indicate if continuously reproducing epaulette sharks are using an income breeding strategy that can be shifted by caloric restriction.

Reproductive hormone changes across the reproductive cycle were minimal (Fig. 4). Female T was elevated in the pre-encapsulating phase (Fig. 4A), where this increase could be inducing egg case secretion from the oviducal glands halfway through the cycle, but the complete picture of endocrine control on the oviducal gland in elasmobranchs is still understudied (Koob and Callard, 1999; Callard et al., 2005). We expected that $P_4$ would peak during the peri-ovulatory and encapsulation period, as $P_4$ is thought to regulate ovulation, egg-retention, and oviposition in other oviparous chondrichthyans (Koob and Callard, 1985; Koob et al.,

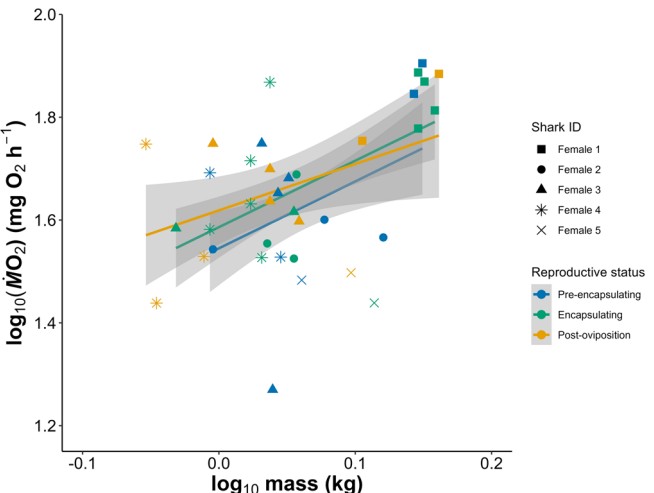

**Fig. 2. The metabolic rate [as RMR (mg $O_2$ h$^{-1}$)] over mass of five female epaulette sharks across three stages of the reproductive cycle.** The linear fits of each reproductive status with individual as a co-variate did not differ between any groupings at $\alpha$=0.05.

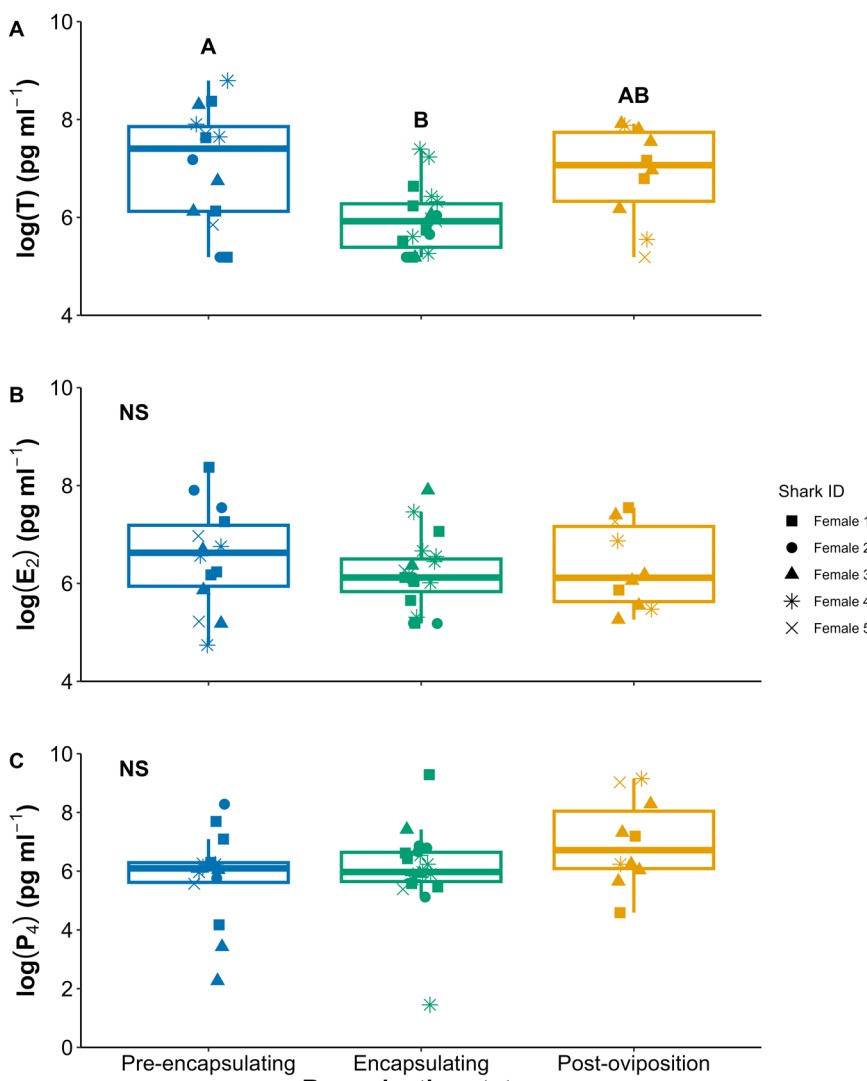

**Fig. 4. Boxplots of the log of circulating reproductive steroid hormone concentrations of (A) testosterone (T), (B) estradiol (E₂), and (C) progesterone (P₄) across female epaulette shark oviparity.** NS denotes non-significance, and differing letters denoted significant differences at $\alpha = 0.05$.

1986; Rasmussen et al., 1999; Awruch et al., 2008; Falahiehzadeh and Salamat, 2020; Inoue et al., 2022). Heupel et al. (1999) serially sampled one captive epaulette shark daily and reported that $P_4$ peaked in the post-oviposition phase. Here, on average, $P_4$ did not change throughout the egg production cycle; however, there were three samples (circled in Fig. S2) during the encapsulating and post-oviposition periods where $P_4$ was 11 to 14 times higher than average (Fig. S2). These could be isolated cases where we captured the short 24–48-h peak in $P_4$ that is easily missed given the brief periodicity. Not including these peaks, our $P_4$ values were similar concentrations to wild reproductively active female epaulette sharks in Heupel et al. (1999). It is still unclear in oviparous chondrichthyans whether the purpose of this mid to late-cycle $P_4$ peak is related to egg-retention and/or oviposition, to stimulate steroidogenesis of T and $E_2$ for the next cycle, and/or to inhibit hepatic synthesis of vitellogenin (Koob and Callard, 1999). Continued experimental work with fine-scale monitoring and hormonal manipulation may help elucidate the undoubtably important role of $P_4$ in chondrichthyan oviparity.

From a hematological standpoint, Hct, [Hb], and MCHC did not differ between reproductive stages. Similarly, in the oviparous cloudy catshark, Hct also did not vary across oviparity (Inoue et al., 2022). To our knowledge, there are no studies that assessed [Hb]

and MCHC across chondrichthyan oviparity, precluding a comparison. Epaulette sharks maintain red blood cell (RBC) size (i.e. no swelling occurs) after air exposure or exhaustive exercise (Schwieterman et al., 2019) but exhibit RBC swelling during anoxia exposure (Chapman and Renshaw, 2009). However, there may also be an effect of captivity on these hematological properties (Wise et al., 1998; Chapman and Renshaw, 2009), and so more research on this species is needed to tease apart hematological responses to cope with various forms of stress versus the influence of reproduction.

Overall, studies explicitly linking routine metabolic rate (RMR) to sex or reproductive state in fishes are limited (Arnold et al., 2021). What little evidence exists suggests that sex- and reproduction-associated differences in metabolic rate can range from weak to strong (Hopkins et al., 1995; Masonjones, 2001; Kraskura et al., 2020; Silva-Garay and Lowe, 2021), but there is no clear understanding of how oviparity versus viviparity influences these costs.

A review of reproductive costs across taxa indicates that ectotherms have lower reproductive costs than endotherms, and that viviparous reproduction is more costly than oviparity (Ginther et al., 2024). Our findings herein support these findings, where epaulette sharks, as oviparous ectotherms, did not demonstrate changes in metabolic rate in relation to the egg-laying cycle.

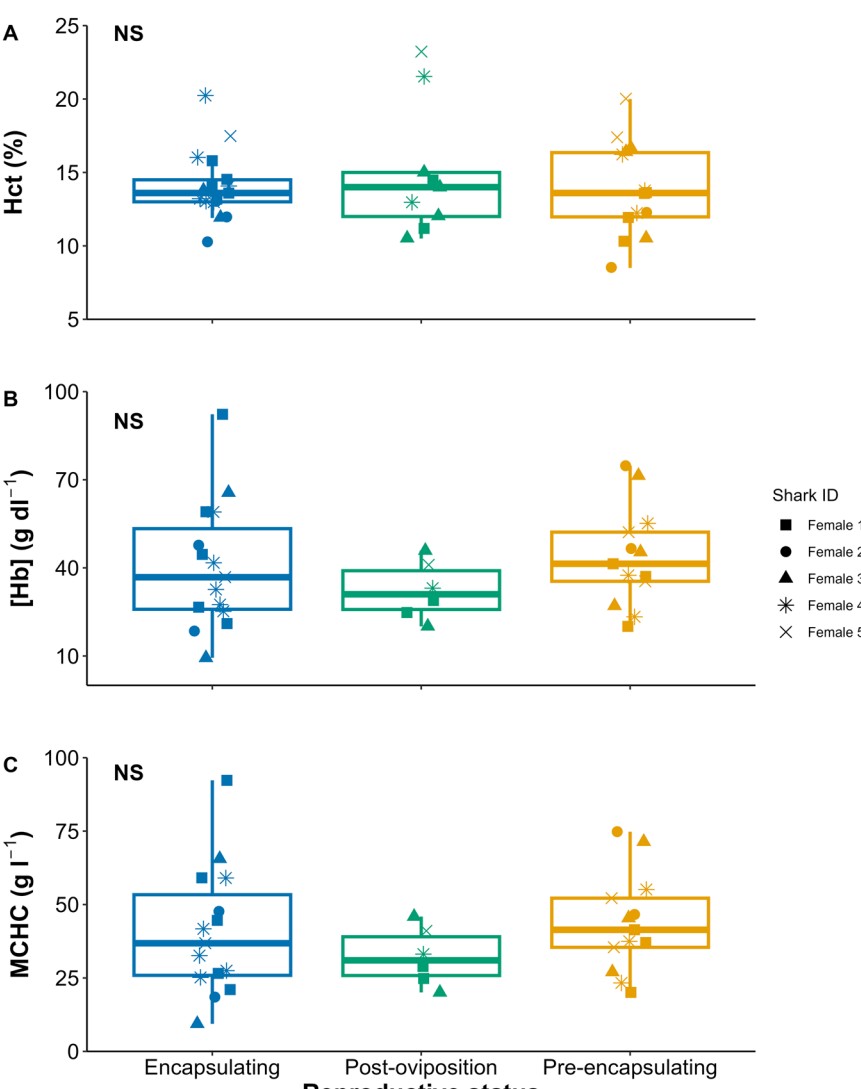

**Fig. 5. Boxplots of the hematological parameters of (A) hematocrit (hct), (B) hemoglobin concentration ([Hb]), and (C) mean corpuscular hemoglobin concentration (MCHC) across female epaulette shark oviparity.** NS denotes non-significance.

Although egg production minimally affected metabolic rate in epaulette sharks, comparisons with non-reproducing individuals may still reveal measurable differences. Indeed, despite the findings herein, direct and indirect reproductive influences on metabolic rates could be substantial in Chondrichthyan fishes and should be carefully considered in study design involving mature individuals. More research that incorporates seasonal variation and a control in the form of a mature but non-reproducing treatment group is needed to further elucidate reproductive effort in the context of an organismal energy budget for Chondrichthyans. This study is the first to our knowledge that directly assesses the effects of oviparity on metabolism in chondrichthyans and has indicated that the reproductive energetic budget is complex and may not be directly reflected in metabolic rate.

## MATERIALS AND METHODS
### Ethics
All experimental protocols in this study were assessed and approved by the James Cook University Animal Ethics Committee (protocol #A2655) and furthermore, conducted in accordance with all relevant guidelines and regulations. Collections were conducted under the appropriate Great Barrier Reef Marine Park Authority (GBRMPA #G19/43380.1) and Queensland Fisheries (200891) permits.

### Animal collection and husbandry
This study included five mature females. Four mature female epaulette sharks over 61 cm total length (TL) (50% maturity at 55 cm TL; Heupel et al., 1999) were hand-collected with dip nets in shallow water from Magnetic Island (n=1, −19.129041, 146.877586) and Balgal Beach, QLD, Australia (n=3, −19.021387, 146.418124) in February and March 2020. Sharks were transported back to the Marine and Aquaculture Research Facility Unit at James Cook University (Douglas, QLD, AUS) within 2 h of capture in 50 L of clean, well-aerated seawater. An additional mature female was sourced from Cairns Marine in September 2020.

Sharks were maintained individually in five 1000-L round tanks connected to a 5500 L reservoir of local natural seawater fitted with a heater, protein skimmer, bio-filtration, and UV sterilization. The system was maintained at 25°C with a half a degree Celsius diel temperature change (i.e. 24.5-25.5°C over a 24-h period) and a 12 h:12 h (light:dark) light cycle to mimic the reproductive seasonality of this species in the wild (Heupel et al., 1999). Water quality parameters (pH, nitrites, nitrates, ammonia) were monitored daily for the first month of introduction to the system and subsequently checked weekly (Table S1). Temperature was monitored within the external reservoir by a sensor that controlled the heater/chiller system as well as individual HOBO pendant loggers (Onset, USA) attached to the standpipe of each tank. Additionally, each tank had one large air stone, a lid constructed of 30% light blocking shade cloth, and a 30 cm×80 cm PVC pipe that was provided as shelter for each shark. All sharks in the study were fed rations at 2% of their body mass three

Biology Open

times weekly (6% body mass per week). Food was comprised of frozen prawn and pilchard, as recommended for small benthic sharks (Janse et al., 2004). Any pieces not consumed were removed, and the amount consumed was recorded. Across the study, sharks consistently consumed all food offered, except for a few instances around oviposition which has been previously documented (Wheeler and Rummer, 2024).

### Female reproductive monitoring

Female reproductive monitoring was conducted from March 2020 until January 2022. During this period, epaulette sharks typically deposited two egg cases per reproductive cycle, with one egg case from each uterus, every 2-4 weeks. We divided the reproductive cycle into three distinct phases: pre-encapsulation, encapsulation of an egg case(s), and post-oviposition (Fig. 1). The post-oviposition phase was defined as the 48-h period following the deposition of the second egg case in a clutch, in cases where eggs were laid across two consecutive nights. This ensured that the full clutch had been deposited before initiating the post-oviposition window. The pre-encapsulation phase lasted several weeks, during which no egg cases were detected via daily palpations. Finally, when egg cases from the next clutch were detected during palpations, the sharks were in the encapsulation phase.

### Respirometry: oxygen uptake measurements

To estimate the routine metabolic rate (RMR) of sharks across reproductive stages, oxygen uptake rates ($\dot{M}O_2$) of each shark were measured using intermittent-flow, static respirometry. The respirometry setup was comprised of opaque PVC chambers with small viewing windows and with baffled ends to allow even water flow and were 15.5 l in volume (15 cm diameter×82 cm length, 1200 l h$^{-1}$ flush recirculating pumps). These chamber sizes proved to be sufficient to obtain linear oxygen uptake slopes while still limiting movement. Oxygen levels within the chambers were measured every two seconds using an OXROB3 fiber optic probe inserted approximately 5 cm into the chamber proper via the overflow outlet connected to a Firesting Optical Oxygen Meter (Pyroscience GmbH, Aachen, Germany).

Sharks were fed 3 days before respirometry trials and then fasted for the next 2 days to ensure the sharks were in a post-absorptive state (Heinrich et al., 2014; Wheeler et al., 2022). Sharks were then carefully introduced into the respirometry chamber at 06:00 h and were constantly supplied with filtered, well-aerated seawater for the first six hours via the flush pump; this period, as determined from preliminary trials, allowed sharks to sufficiently habituate to the chambers following the transfer. At 12:00 h, a relay timer was used to intermittently turn off the flush pump for 5 min. These time intervals were long enough to ensure that the decline in O$_2$ was linear (average $R^2$=0.96) but short enough such that O$_2$ levels within chambers did not decrease below 80% saturation at any point of the trials. Following each of the O$_2$ uptake measurement periods, the flush pump was turned on once again, thus returning O$_2$ levels in the chamber water back to 100% air saturation; flush duration was ten minutes. These measurements and flush cycles were repeated for 24 h until 12:00 h of the following day to ensure sufficient data points were collected for each individual. Immediately after the shark was removed from the respirometry chamber, the shark was weighed (i.e. wet mass in grams).

Oxygen uptake rates ($\dot{M}O_2$ in O$_2$ h$^{-1}$) were calculated using the *RespiroRS* package in R (Merciere and Norin, 2021). To estimate RMR, we used the lowest 10% of $\dot{M}O_2$ measurements recorded across each 24-h respirometry trial (Chabot et al., 2016). Given that all individuals in this study were actively reproducing, we deliberately use the term routine metabolic rate rather than standard metabolic rate, as the latter assumes a non-reproductive state (Chabot et al., 2016). This method allowed us to capture the lowest sustained metabolic output while accounting for diel variation, which is known to peak at night in epaulette sharks (Wheeler et al., 2022). Although calculations demonstrated microbial respiration to be negligible (less than 5% of shark respiration), each respirometry chamber was cycled empty for 30 min before and after each trial to account for microbial respiration.

### Blood collection and analyses

Immediately following each respirometry trial at 12:00 h, the shark was removed and a small blood sample (1.5–2 ml, not exceeding 0.2% of the shark's body mass) was collected via caudal puncture using a 23 G×1″ needle and a syringe coated with sodium heparin in a phosphate buffered saline. After the sampling, sharks were returned to their respective holding condition, where they typically resumed feeding within three hours and always within 24 h. To minimize stress and avoid affecting subsequent reproductive cycles, blood sampling was conducted at intervals of at least two weeks. All samples were stored on ice for no more than 15 min, after which time, three hematological metrics that reflect blood-oxygen transport capacity were assessed. Hematocrit (Hct, in %), was measured as the ratio of packed red blood cells to total blood volume via capillary centrifuging at 11,500 rpm (13,000 $g$) for 5 min. Hemoglobin ([Hb] in g dl$^{-1}$) was measured using a HemoCue (Brae, CA) that has been previously validated for epaulette sharks (i.e. applying a correction factor [Hb]*0.91–0.53; Schwieterman et al., 2019). Mean corpuscular hemoglobin concentration (MCHC, g l$^{-1}$) was calculated by dividing [Hb] by Hct then multiplying by 100. Subsequently, whole blood samples were centrifuged at 1240 $g$ for 5 min, and the plasma was removed and stored at −80°C for later steroid hormone analyses.

The reproductive steroid hormones testosterone (T), 17β-estradiol (E$_2$), and progesterone (P$_4$) were measured using radioimmunoassay (RIA), following protocols described in Awruch et al. (2008). First, each plasma sample was extracted twice from 300 μl of plasma using 1500 μl of ethyl acetate (1:5) with 30 s of vortexing and subsequent freezing at −20°C before the aqueous phase was separated. After the second extraction, duplicate 100 μl aliquots were evaporated into assay tubes before reconstitution with a 0.05 M phosphate buffer (0.1% gelatin, 0.01% Thimerosal). To account for procedural loss, five replicates of pooled sample were spiked with 3000 counts min$^{-1}$ of H$^3$-labelled steroids (Perkin-Elmer, Australia) and extracted twice in the same manner as the samples. The extraction efficiencies were determined to be 91, 84, and 83% for T, E$_2$, and P$_4,$ respectively. The antisera for the assays were obtained from Novus Biologicals (Australia), and assays were validated by parallelism to serially diluted standard curves. All radioactivity was detected with a Hidex 300 SL liquid scintillation counter (Turku, Finland). The lower detection limits of the assay were 68.2, 65.6, and 30.7 pg ml$^{-1}$ for T, E$_2$, and P$_4$, respectively. The intra-assay variances were 8, 10, and 9%, and the inter-assay variances were 9, 11, and 12% for T, E$_2$, and P$_4,$ respectively.

### Statistical analyses

The $\dot{M}O_2$ data met the assumptions of normality (after log transformation) and homoscedasticity, and the hormone concentrations and hematological parameters that were used as dependent variables were log transformed to achieve normality. All data were assessed across the three discrete female reproductive stages (Fig. 1). For $\dot{M}O_2$ data, a linear mixed-effects model was fitted using log-transformed mass and reproductive status as fixed effects, with individual (ID) included as a random intercept. This approach accounts for both variation in body size and repeated measures across individuals. For all hormone and hematological parameters, ANCOVAs with individual as a covariate were conducted. The *emmeans* package (Lenth, 2022) was used for *post hoc* comparisons between reproductive status categories when applicable. All statistical analyses were conducted in R (version 4.5.1, R Core Development Team, 2023), where results were considered significant at $\alpha$=0.05. All resulting boxplots show the interquartile range (25th to 75th percentiles), the horizontal line marks the median, whiskers extend to the most extreme values within the 1.5 multiple of the interquartile range, and any points beyond this range are shown as outliers.

### Acknowledgements

The authors would like to thank past and present Rummer Lab members for their assistance with husbandry and data collection during this project. Furthermore, we thank S. Wever and B. Lawes of the Marine Aquaculture Research Facilities Unit at JCU for their technical support. An anonymous donor to the Anderson Cabot Center at the New England Aquarium helped fund key equipment used in this study. Finally, we thank J. Kneebone, two anonymous examiners of C.R.W.'s PhD dissertation, and two anonymous reviewers of this manuscript for their invaluable feedback.

### Competing interests

The authors declare no competing or financial interests.

## Author contributions

Conceptualization: C.R.W., J.W.M., J.L.R.; Data curation: C.R.W.; Formal analysis: C.R.W., C.A.A.; Funding acquisition: C.R.W., C.A.A., J.W.M., J.L.R.; Investigation: C.R.W., C.A.A., J.W.M., J.L.R.; Methodology: C.R.W., C.A.A., J.W.M., J.L.R.; Project administration: C.R.W., J.W.M., J.L.R.; Resources: C.A.A., J.W.M., J.L.R.; Supervision: C.A.A., J.W.M., J.L.R.; Validation: C.R.W.; Visualization: C.R.W.; Writing – original draft: C.R.W.; Writing – review & editing: C.R.W., C.A.A., J.W.M., J.L.R.

## Funding

Support for this work came from an American Australian Association graduate scholarship, a JCU Postgraduate Research Scholarship, the Australian Research Council Centre of Excellence for Coral Reef Studies, The Australian Wildlife Society, and the Australian Society for Fish Biology. Open Access funding provided by James Cook University. Deposited in PMC for immediate release.

## Data and resource availability

All relevant data and details of resources can be found within the article and its supplementary information. Data are also openly available from Research Data JCU at James Cook University at doi:10.25903/s8ws-c294.

## Peer review history

The peer review history is available online at https://journals.biologists.com/bio/lookup/doi/10.1242/bio.062076.reviewer-comments.pdf

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
