## [Peer Review File · Biology Open]

Assessing the metabolic and physiological costs of oviparity in the epaulette shark (*Hemiscyllium ocellatum*)

Cynthia Awruch, John Mandelman, Jodie Rummer and Carolyn Wheeler

DOI: 10.1242/bio.062076

Editor: Lewis Halsey

Review timeline

Original submission:	18 May 2025
Editorial decision:	23 May 2025
First revision received:	5 September 2025
Editorial decision:	5 September 2025
Second revision received:	2 October 2025
Accepted:	2 October 2025

Original submission

First decision letter

MS ID#: bio.062076

MS Title: Assessing the metabolic and physiological costs of oviparity in the epaulette shark (*Hemiscyllium ocellatum*)

Authors: Carolyn Wheeler; Cynthia Awruch; John Mandelman; Jodie Rummer

Dear Dr Wheeler,

I have now reached a decision on the above manuscript.

The reviewer reports are shown at the bottom of this email or can be accessed, together with a copy of this decision letter, by going to:

As you will see, the reviewers gave favourable reports, but raised some critical points that will require amendments to your manuscript. I hope that you will be able to carry these out, because we would like to be able to accept your paper.

Reviewer 1

Comments to Author

In general I found this to be an interesting and well written study, and an important contribution to the field. My comments relatively minor:

Maybe worth mentioning something about the water (local sea water or artificial). By the way, the water parameters are in Table S1 (Not S4.1).

Line 179. I am a bit confused by the statement that "only between the hours of 0900-1500hr were used" when considering that the measurements started at 1200hr and ended at 1200hr the next

day. Does it mean that data from 1200-1500hr from day one and from 0900-1200hr from day two was used?

Lines 230, 234, 235: Should be Table 1 (not 4.1). Actually, please check all Table and Figure numbering as more seems to be wrong.

Line 146. Food consumption was recorded but this data is not shown. Was it that some feed was always left over, suggesting that they were fed ad libitum? That would be important to know since if they were not food restricted then they had all energy they could handle for growth and reproduction and the lack of effects on the reproductive cycle on metabolic rate could reflect a relocation of energy from growth to reproduction when needed.

Here is a rather serious thought that the authors may consider (or at least mention this possibility and arguments against it): Could it be that the starvation period used before respirometry pushed the shark's metabolism down to a minimum maintenance level where both investments in growth and reproduction was temporarily stopped? That would explain why no effect of reproductive state was seen: the sharks were just halting everything while waiting for food. If the authors have data on reproductive output (egg production) from periods where no respirometry was done, that could indicate if the starvation periods linked to respirometry slowed down the reproductive effort.

Reviewer 2

Comments to Author

This study investigated the metabolic cost of egg production in tropical epaulette sharks. It supplemented measurements of metabolic rate (oxygen uptake rate) with measurements of reproductive hormones and blood parameters (haematocrit, haemoglobin concentration). The motivation for the study was that little is known about metabolic costs of reproduction in chondrichthyan fishes. Results are primarily negative, showing mostly no detectable costs to egg production or physiological changes across the reproductive cycle.

The manuscript is well written and well structured, and the experiments appear to be well conducted. I think the authors have done an overall great job with both the study and manuscript. I do have a series of comments below, but I think these should be manageable to address.

I guess the sample size of five individual sharks (with each individual measured multiple times) is on the low side. I have not worked with sharks myself, and even though these are small sharks, I can imagine that there could be logistical, legislative, and/or ethical issues preventing a larger sample size, so I can sympathise with this. I also do not see any signs that the authors would have reached other conclusion with a larger sample size, although I did wonder about two things in particular related to sample size that could / should be clarified in the manuscript.

First, it is a bit unclear how many repeated measurements were actually done on the five sharks, and why they were distributed across individuals as they were. Line 241 states that "Respirometry was performed 37 times throughout this study". Figure S4.1 shows the distribution of repeated MO₂ measurements per individual, but there are 40 measurements (data points) in the figure, not 37. The number of measurements per individual is also very uneven. The discrepancy between 37 and 40 measurements should be cleared up, and I think an explanation for why the number of MO₂ measurements is so unevenly distributed between individual ought to be included in the main manuscript.

Second, the methods sections states that four sharks were wild caught while the fifth was from Cairns Marine (line 133), a company supplying the aquarium industry, but there is no mention of which individual this fifth shark is. I noticed from both Table 1 and Figure 2 that 'Female 5' appear to stand out from the remaining four individuals in both reproductive data (Table 1) and metabolic rate for its size (consistently low; Figure 2). Is Female 5 the individual from Cairns Marine? Any explanation for why it appears to deviate from the wild-caught individuals? Is it an issue?

I think the intermittent-flow respirometry protocol employed here is good and solid, but it is unclear why the authors focus on routine metabolic rate, and what actually went into its calculation. First, the authors measured MO₂ once every 15 min over 24 h (lines 171-172 and 175-177), which must have produced ~96 MO₂ measurements per individual per respirometry trial. This number of MO₂ measurements is sufficient for estimating standard (or resting) metabolic rate, which is often calculated as the mean of the lowest 10% (or some quantile) of the MO₂ values. The authors did not do this but instead chose to discard all measurements before 9:00 in the morning and after 15:00 in the afternoon. From these ~24 MO₂ measurements, they have calculated what they call routine metabolic rate, presumably as an overall average of the ~24 measurements, although this is unclear. It needs to be specified how routine metabolic rate was calculated. I also think the authors ought to mention/justify why they chose to calculate routine metabolic rate rather than some resting value, given that routine metabolic rate implies that some (unknown) component of physical activity is included in the measurement, which could perhaps have masked any differences in MO₂ across the reproductive cycle, if the unquantified activity changed with reproductive status. I also think it should be justified why the authors discarded all measurements outside of 9:00-15:00, rather than simply using all data and taking an average of the lowest X% as a measure of resting metabolic rate.

Another general comment is that the abstract states that metabolic rate scaled with body mass (lines 45-46), but Table S2 indicates that mass did not have a significant effect on metabolic rate (MO₂) ($p = 0.39$), in which case metabolic rate did not actually scale with body mass. This should be clarified. Another issue here is that Table S2 indicates that the relationship between MO₂ and both mass and reproductive status was analysed in a simple linear model with raw data. It is well known that metabolic rate most often scales allometric (disproportionally) with body mass according to a power function, and thus the relationship between MO₂ and body mass is most likely not linear, and both MO₂ and body mass should arguably be log-transformed. Also, given that individuals had between 3 and 12 repeated measurements of MO₂, according to Figure S4.1, I would argue that shark ID should be included in the model as a random effect, rather than a fixed effect as it appears to be from Table S2. That is, the model should arguably be $\text{lmer}(\log_{10}(\text{MO}_2) \sim \log_{10}(\text{mass}) + \text{reproductive status} + (1 | \text{ID}))$, with the interaction (*) dropped if non-significant.

Finally, the authors indicate with their motivation for the study that we know more about the energetic costs of reproduction in other fishes (e.g., teleosts), but there is no mention or discussion of what is known about costs of reproduction in these other fishes. This ought to be included in the manuscript.

Specific comments:

Lines 74-76: Ginther et al. (2024) would seem like a good reference for the statement here that "The degree of reproductive effort is inherently linked to the reproductive strategy, with viviparity (live-bearing) generally requiring more sustained effort than oviparity (egg-laying) due to the extended duration and maintenance demands associated with gestation."

Lines 78-80: The authors write here that "In other vertebrates, such as reptiles and passerine birds, reproductive costs ..." but it is unclear what 'other' refers to.

Lines 86-87: The authors state here that "This study quantified fine-scale changes in the metabolic rates associated with reproduction in chondrichthyan fishes (sharks, rays, skates and chimaeras)." This is an overstatement or typo, as 'fishes' means multiple species, and this study used only one.

Lines 151-152: I am a little confused about the statement that "The post-oviposition phase was defined as the first 48 hours following the deposition (laying) of a clutch." It is stated in line 235 that eggs could be laid up to two nights apart, but it is unclear how this variability is considered in the 48 h post-oviposition phase.

Line 202: It would be good to write out what 'T', 'E2', and 'P4' are.

Lines 264-272: Do you think your results had been different if eggs had been fertilised by a male? Is there any evidence to support or reject this that could be discussed here (or elsewhere)?

Line 325: It is unclear what the 'differences' here refer to. Differences in what?

Lines 512-515 (Table 1): Table legend is a bit messy here.

Lines 538-540 (Figure 2): I think it would be nice to add here how many times each shark was measured for MO2. Perhaps in parenthesis after each female in the 'Shark ID' legend.

Lines 546-547 (Figure 3) and elsewhere: It ought to be included what the boxes and their limits/lines represent in boxplots.

Reviewer's Responses to Questions

Experimental quality

Does each figure have the proper controls?

If 'No', please indicate reasons in Comments for Author box below.

Reviewer #1:

- Yes

Reviewer #2:

- Yes

Were the data analyzed using appropriate statistical tests?

If 'No', please indicate reasons in Comments for Author box below.

Reviewer #1:

- Yes

Reviewer #2:

- No

Reproducibility

Were experiments performed using adequate number of biological replicates?

If 'No', please indicate reasons in Comments for Author box below.

Reviewer #1:

- Yes

Reviewer #2:

- Yes

Does the methods section provide sufficient detail to permit reproducibility?

If 'No', please indicate reasons in Comments for Author box below.

Reviewer #1:

- Yes

Reviewer #2:

- Yes

Completeness

Are the manuscript's conclusions supported by the data?

If 'No', please indicate reasons in Comments for Author box below.

Reviewer #1:

- Yes

Reviewer #2:

- Yes

Scholarship

Do the authors cite and discuss the merits of data that would argue for and against their conclusion?

If 'No', please indicate reasons in Comments for Author box below.

Reviewer #1:

- Yes

Reviewer #2:

- Yes

Does the manuscript title & abstract accurately reflect the contents of the manuscript, without hyperbole?

If 'No', please indicate reasons in Comments for Author box below.

Reviewer #1:

- Yes

Reviewer #2:

- Yes

First revision**Author response to reviewers' comments****Reviewer 1**

Maybe worth mentioning something about the water (local sea water or artificial). By the way, the water parameters are in Table S1 (Not S4.1).

Water in system was local seawater, added in line 135. Table S1 in-text reference amended.

Line 179. I am a bit confused by the statement that "only between the hours of 0900-1500hr were used" when considering that the measurements started at 1200hr and ended at 1200hr the next day. Does it mean that data from 1200-1500hr from day one and from 0900-1200hr from day two was used?

This approach has now been replaced with a 10th percentile approach to calculating RMR, see reviewer 2 comments and responses below.

Lines 230, 234, 235: Should be Table 1 (not 4.1). Actually, please check all Table and Figure numbering as more seems to be wrong.

Checked throughout MS. These 4.X numberings are left over from thesis formatting.

Line 146. Food consumption was recorded but this data is not shown. Was it that some feed was always left over, suggesting that they were fed ad libitum? That would be important to know since if they were not food restricted then they had all energy they could handle for growth and reproduction and the lack of effects on the reproductive cycle on metabolic rate could reflect a relocation of energy from growth to reproduction when needed.

Sharks were fed 3% of their body weight three times per week, and any uneaten food was collected and weighed at the end of each feeding session. Across the study, sharks consistently consumed all food offered, with the exception of a few instances around oviposition. These observations suggest that the animals were not food-restricted and had access to sufficient energy for both growth and reproduction.

We also considered whether the fasting period prior to respirometry might interfere with natural egg-laying patterns. However, this did not appear to be the case. In several instances, females were identified via palpation as having eggs in utero, and fasting was initiated with the intent of capturing metabolic rate during encapsulation. Despite this, some females deposited their eggs during the fasting period, before respirometry could be conducted. This indicates that egg deposition was not dependent on recent feeding and that fasting did not energetically delay oviposition. Discussion of this has been added at lines 149-150.

Here is a rather serious thought that the authors may consider (or at least mention this possibility and arguments against it): Could it be that the starvation period used before respirometry pushed the shark's metabolism down to a minimum maintenance level where both investments in growth and reproduction was temporarily stopped? That would explain why no effect of reproductive state was seen: the sharks were just halting everything while waiting for food. If the authors have data on reproductive output (egg production) from periods where no respirometry was done, that could indicate if the starvation periods linked to respirometry slowed down the reproductive effort.

This is an important point, and we have added a brief discussion of this possibility to the manuscript now at lines 282-292.

While it is theoretically plausible that fasting prior to respirometry could suppress metabolic activity to a maintenance level and temporarily halt energetic investment in growth or reproduction, our observations suggest this was not the case. In several instances, females were palpated and confirmed to have eggs in utero, and fasting was initiated with the intent of capturing metabolic rate during encapsulation. Despite this, many of these females deposited their eggs during the fasting period, before respirometry could be conducted. This indicates that egg deposition proceeded even in the absence of recent feeding, suggesting that the energetic demands of reproduction were met without interruption.

Furthermore, reproductive output was monitored throughout the study, including during periods when respirometry was not performed. We observed no consistent delays or reductions in egg production associated with fasting, and oviposition occurred regularly across all reproductive stages. These findings support the interpretation that the fasting protocol did not suppress reproductive effort and that the lack of a metabolic signal across reproductive stages reflects true physiological stability rather than an artifact of energetic shutdown.

We have clarified these points in the revised discussion to address this concern directly.

Reviewer 2

I guess the sample size of five individual sharks (with each individual measured multiple times) is on the low side. I have not worked with sharks myself, and even though these are small sharks, I can imagine that there could be logistical, legislative, and/or ethical issues preventing a larger sample size, so I can sympathise with this. I also do not see any signs that the authors would have reached other conclusion with a larger sample size, although I did wonder about two things in particular related to sample size that could / should be clarified in the manuscript.

We agree that a larger sample size would have strengthened our conclusions, and this was our original goal. However, field collections coincided with the onset of the COVID-19 pandemic (March 2020), which introduced unanticipated logistical challenges and travel restrictions that prevented us from sourcing additional females from other reef areas. Permit conditions from the Great Barrier Reef Marine Park Authority also limited the number of females we could collect per region, and follow-up collection trips yielded only large males (90-100 cm total length). The sole female was obtained from Cairns Marine after extensive searches. These combined factors constrained our final sample size.

First, it is a bit unclear how many repeated measurements were actually done on the five sharks, and why they were distributed across individuals as they were. Line 241 states that "Respirometry was performed 37 times throughout this study". Figure S4.1 shows the distribution of repeated MO2 measurements per individual, but there are 40 measurements (data points) in the figure, not 37. The number of measurements per individual is also very uneven. The discrepancy between 37 and

40 measurements should be cleared up, and I think an explanation for why the number of MO₂ measurements is so unevenly distributed between individual ought to be included in the main manuscript.

We thank the reviewer for pointing out the discrepancy between the number of respirometry trials reported in the manuscript and the number of data points shown in Figure S4.1 (now S1). The correct number of trials conducted was 37, as stated in the main text. The original version of Figure S4.1 (now S1) inadvertently included three additional data points due to a filtering error in the plotting script. This has now been corrected, and the updated figure accurately reflects the 37 trials. This filtering error only applied to this figure and none of the other models or figures. The uneven distribution of respirometry trials across individuals was largely due to logistical challenges associated with reproductive timing. In many instances, females were identified via palpation as having eggs in utero, prompting the start of a fasting period in preparation for respirometry. However, several of these females deposited their eggs during the fasting window, prior to the scheduled trial, which meant they could no longer be sampled at the intended encapsulating stage. As a result, while trials were initiated with the goal of capturing metabolic rate during encapsulation, the reproductive status of some individuals shifted before measurements could be taken. Additionally, it is worth noting that four of the five females in this study were also part of a separate investigation, and their sampling schedules were coordinated across both projects. Female 5 was added later and sampled independently, which further contributed to the variation in trial numbers.

I think the intermittent-flow respirometry protocol employed here is good and solid, but it is unclear why the authors focus on routine metabolic rate, and what actually went into its calculation. First, the authors measured MO₂ once every 15 min over 24 h (lines 171-172 and 175-177), which must have produced ~96 MO₂ measurements per individual per respirometry trial. This number of MO₂ measurements is sufficient for estimating standard (or resting) metabolic rate, which is often calculated as the mean of the lowest 10% (or some quantile) of the MO₂ values. The authors did not do this but instead chose to discard all measurements before 9:00 in the morning and after 15:00 in the afternoon. From these ~24 MO₂ measurements, they have calculated what they call routine metabolic rate, presumably as an overall average of the ~24 measurements, although this is unclear. It needs to be specified how routine metabolic rate was calculated. I also think the authors ought to mention/justify why they chose to calculate routine metabolic rate rather than some resting value, given that routine metabolic rate implies that some (unknown) component of physical activity is included in the measurement, which could perhaps have masked any differences in MO₂ across the reproductive cycle, if the unquantified activity changed with reproductive status. I also think it should be justified why the authors discarded all measurements outside of 9:00-15:00, rather than simply using all data and taking an average of the lowest X% as a measure of resting metabolic rate.

Our decision to focus on routine metabolic rate was based on both biological and methodological considerations. Specifically, all individuals in this study were actively reproducing, and as such, do not meet the criteria for estimating standard metabolic rate, which requires animals to be post-absorptive, inactive, and non-reproductive. For this reason, we deliberately use the term routine metabolic rate, which reflects baseline metabolism inclusive of low-level activity and physiological processes associated with reproduction.

Regarding the calculation, we selected MO₂ measurements collected between 09:00 and 15:00 based on prior work (Wheeler et al. 2022) using the same individuals, which demonstrated a strong circadian rhythm in metabolic rate. MO₂ values were consistently lowest during this daytime window, and excluding measurements outside this period helped minimize the influence of nocturnal activity and feeding-related metabolic spikes. This approach was intended to approximate resting conditions as closely as possible within the constraints of a reproductive study. To address the reviewer's suggestion directly, we recalculated routine metabolic rate using the lowest 10% of MO₂ values across the full 24-hour period for each trial. These revised estimates were consistent with those derived from our original 09:00-15:00 window, confirming that our initial approach effectively captured the lowest metabolic rates. Given this alignment, we have updated the manuscript to reflect the use of the 10% quantile method as our final metric for routine metabolic rate, and clarified this in the methods section accordingly.

Another general comment is that the abstract states that metabolic rate scaled with body mass (lines 45-46), but Table S2 indicates that mass did not have a significant effect on metabolic rate (MO₂) ($p = 0.39$), in which case metabolic rate did not actually scale with body mass. This should be clarified. Another issue here is that Table S2 indicates that the relationship between MO₂ and both mass and reproductive status was analysed in a simple linear model with raw data. It is well known that metabolic rate most often scales allometric (disproportionally) with body mass according to a power function, and thus the relationship between MO₂ and body mass is most likely not linear, and both MO₂ and body mass should arguably be log-transformed. Also, given that individuals had between 3 and 12 repeated measurements of MO₂, according to Figure S4.1, I would argue that shark ID should be included in the model as a random effect, rather than a fixed effect as it appears to be from Table S2.

That is, the model should arguably be $\text{lmer}(\log_{10}(\text{MO}_2) \sim \log_{10}(\text{mass}) * \text{reproductive status} + (1 | \text{ID}))$, with the interaction (*) dropped if non-significant.

We agree that the initial analysis presented in Table S2 did not adequately reflect the expected allometric scaling of metabolic rate with body mass, nor did it appropriately account for repeated measures within individuals.

In response, we have revised our statistical approach to better align with established metabolic theory and the structure of our data. Specifically:

1. Log-transformation of variables: We now model the relationship between metabolic rate and body mass using log₁₀-transformed values for both MO₂ and mass, consistent with the expectation of a power-law (allometric) relationship. This transformation improves model fit and interpretability, and aligns with standard practice in metabolic scaling studies.
2. Mixed-effects modeling: To account for repeated measurements per individual (as shown in Figure S4.1), we have implemented a linear mixed-effects model with shark ID included as a random intercept. This allows us to account for individual-level variation in baseline metabolic rate while estimating population-level effects of mass and reproductive status.
3. Model selection and interpretation: Our final model is specified as

We tested the inclusion of an interaction term between mass and reproductive status, but it was not statistically significant and did not improve model fit ($\Delta\text{AIC} = 3.67$, $p = 0.85$), so it was excluded from the final model.

While the effect of mass was not statistically significant in this final model ($p = 0.32$), the direction of the relationship was positive and consistent with theoretical expectations. We have revised the abstract to clarify that although metabolic rate scaled positively with body mass, this relationship was not statistically significant in our dataset.

Finally, the authors indicate with their motivation for the study that we know more about the energetic costs of reproduction in other fishes (e.g., teleosts), but there is no mention or discussion of what is known about costs of reproduction in these other fishes. This ought to be included in the manuscript.

We have added a brief discussion of what is known about reproductive costs in other fishes (e.g., teleosts) in the revised manuscript (Lines 343-349). We note, however, that the available data remain limited—few studies directly quantify routine metabolic rate changes across reproductive stages in fish—which underscores the novelty and importance of our work in chondrichthyans.

Lines 74-76: Ginther et al. (2024) would seem like a good reference for the statement here that "The degree of reproductive effort is inherently linked to the reproductive strategy, with viviparity (live-bearing) generally requiring more sustained effort than oviparity (egg-laying) due to the extended duration and maintenance demands associated with gestation."

The authors agree and this reference had been added in line 76-77.

Lines 78-80: The authors write here that "In other vertebrates, such as reptiles and passerine birds, reproductive costs ..." but it is unclear what 'other' refers to.

This is a good point given that fish have not yet been introduced. Amended in text to "In vertebrates such as..."

Lines 86-87: The authors state here that "This study quantified fine-scale changes in the metabolic rates associated with reproduction in chondrichthyan fishes (sharks, rays, skates and chimaeras)." This is an overstatement or typo, as 'fishes' means multiple species, and this study used only one.

Agreed, changed to "in a chondrichthyan fish (class of fishes including sharks, rays, skates and chimaeras)."

Lines 151-152: I am a little confused about the statement that "The post-oviposition phase was defined as the first 48 hours following the deposition (laying) of a clutch." It is stated in line 235 that eggs could be laid up to two nights apart, but it is unclear how this variability is considered in the 48 h post-oviposition phase.

To clarify, the post-oviposition phase was defined as the 48-hour period following the deposition of the second egg in a clutch, in cases where eggs were laid across two consecutive nights. This approach ensured that the full clutch had been deposited before initiating the post-oviposition window, thereby maintaining consistency in how reproductive phases were assigned. We have updated the methods section to reflect this clarification.

Line 202: It would be good to write out what 'T', 'E2', and 'P4' are.

Agreed, hormones now spelled out in full at first appearance, now line 204.

Lines 264-272: Do you think your results had been different if eggs had been fertilised by a male? Is there any evidence to support or reject this that could be discussed here (or elsewhere)?

To our knowledge, there is no published information on whether fertilization affects the energetic cost of egg production in oviparous elasmobranchs. However, based on our observations, the size and quality of eggs produced by females in this study were consistent with those produced later when males were introduced as part of a breeding program. This suggests that, from an energetic standpoint, fertilized and unfertilized eggs are comparable in terms of yolk investment. Additionally, because embryos in epaulette sharks are macroscopic at the time of egg deposition, they are not metabolically active within the female and would not contribute to measurable oxygen consumption during respirometry. We have added a brief note to the discussion to clarify this point.

Line 325: It is unclear what the 'differences' here refer to. Differences in what?

Sentence rewritten at lines 353-355 as: "Although egg production minimally affected metabolic rate in epaulette sharks, comparisons with non-reproducing individuals may still reveal measurable differences."

Lines 512-515 (Table 1): Table legend is a bit messy here.

We appreciate the recommendation to streamline the legend for Table 1. However, the current phrasing is necessary to define each reproductive and morphological metric in a single, self-contained statement. Removing any component risks forcing readers to search the main text for

definitions, which could interrupt the flow of interpretation. For these reasons, we have elected to retain the existing legend as originally submitted.

Lines 538-540 (Figure 2): I think it would be nice to add here how many times each shark was measured for MO₂. Perhaps in parenthesis after each female in the 'Shark ID' legend.

These counts are provided in Supplementary Figure S1 to avoid overloading the main figure legend and maintain clarity. We believe referring readers to the supplemental material ensures the information remains accessible without cluttering the primary presentation.

Lines 546-547 (Figure 3) and elsewhere: It ought to be included what the boxes and their limits/lines represent in boxplots.

A sentence has been added at lines 237-239 that describes the components of the boxplots in all figures.

Second decision letter

MS ID#: bio.062076R1

MS Title: Assessing the metabolic and physiological costs of oviparity in the epaulette shark (*Hemiscyllium ocellatum*)

Authors: Carolyn Wheeler; Cynthia Awruch; John Mandelman; Jodie Rummer

Dear Dr Wheeler,

I've had a chance to read your rebuttal and your manuscript changes pretty much straight away.

Overall I feel you've done a good job of attending to the Reviewers' concerns. However, I would like it to be explicitly recognised in the Abstract that the sample size is five. I appreciate working with sharks is hard, and 5 is a great deal more information than 0. However, particularly given you have a number of 'null' results (which are totally fine), a low sample size makes it that much harder to be sure about interpreting findings. Put simply, with standard statistics it's hard to demonstrate no effect, particularly with a small sample size. Please therefore also include the would tentative(ly) once in the Abstract.

That's all.

Third decision letter

MS ID#: bio.062076R2

MS Title: Assessing the metabolic and physiological costs of oviparity in the epaulette shark (*Hemiscyllium ocellatum*)

Authors: Carolyn Wheeler; Cynthia Awruch; John Mandelman; Jodie Rummer

Dear Dr Wheeler,

I've read through your thorough Reviewer response. I am happy to tell you that your manuscript has now been accepted for publication in Biology Open, pending our standard publication integrity checks. It was accepted on 2nd October 2025.